# Development, Optimisation and Validation of a Novel Multiplex Real-Time PCR Method for the Simultaneous Detection of *Cryptosporidium* spp., *Giardia duodenalis* and *Dientamoeba fragilis*

**DOI:** 10.3390/pathogens11111277

**Published:** 2022-10-31

**Authors:** Isbene Sánchez, Alejandro Dashti, Pamela C. Köster, Begoña Bailo, Nuria González, Janire Allende, Christen Rune Stensvold, David Carmena, David González-Barrio

**Affiliations:** 1Vacunek SL, Bizkaia Technology Park, 48160 Derio, Spain; 2Parasitology Reference and Research Laboratory, Spanish National Centre for Microbiology, Health Institute Carlos III, Majadahonda, 28220 Madrid, Spain; 3Department of Bacteria, Parasites and Fungi, Infectious Disease Preparedness, Statens Serum Institute, 2300 Copenhagen, Denmark; 4Center for Biomedical Research Network (CIBER) in Infectious Diseases, Health Institute Carlos III, Majadahonda, 28220 Madrid, Spain

**Keywords:** diagnosis, enteric parasites, diarrhoea, clinical setting

## Abstract

The enteric protozoan parasites *Cryptosporidium* spp., *Giardia duodenalis* and *Dientamoeba fragilis* are—to various extents—contributors to the burden of gastrointestinal illness in high-income countries. Detection of these pathogens by microscopy examination is challenging because of the limited sensitivity and need for specific staining procedures. We developed and optimised a new multiplex real-time PCR assay for the simultaneous detection of *Cryptosporidium* spp., *G. duodenalis* and *D. fragilis* in clinical (stool) samples. The diagnostic performance of the assay was evaluated against a large panel of well-characterised DNA samples positive for *Cryptosporidium* spp. (*n* = 126), *G. duodenalis* (*n* = 132) and *D. fragilis* (*n* = 49). The specificity of the test was assessed against a DNA panel from other intestinal or phylogenetically related parasites (*n* = 105) and faecal DNA from individuals without clinical manifestations (*n* = 12). The assay exhibited a diagnostic sensitivity of 0.90–0.97 and a diagnostic specificity of 1. The limit of detection was estimated for *Cryptosporidium* (1 oocyst) and *G. duodenalis* (5 × 10^−4^ cysts). The method allowed the detection of four *Cryptosporidium* species (*C. hominis*, *C. parvum*, *C. meleagridis* and *C. cuniculus*) and five *G. duodenalis* assemblages (A–E) without cross-reacting with other parasites belonging to the phyla Amoebozoa, Apicomplexa, Euglenozoa, Microsporidia, Nematoda and Platyhelminthes. This newly developed multiplex real-time PCR assay represents a novel alternative for the rapid and accurate detection of *Cryptosporidium*, *G. duodenalis* and *D. fragilis* in clinical settings.

## 1. Introduction

Diarrhoea is a major health problem worldwide. Only in 2017, almost 1.6 million people died from diarrheal diseases globally (https://ourworldindata.org/diarrheal-diseases#, accessed on 7 October 2022). Those most affected by diarrhoea are children and immunocompromised individuals living in developing countries with limited or no access to safe water and adequate sanitation. Even though diarrhoea-related mortality rates in resource-poor settings have decreased considerably, morbidity remains high [1]. In most instances, the aetiologies of diarrhoea are related to viral, bacterial and parasitic infections, differing in severity and public health impact according to endemic level and geographical area.

Among parasitic pathogens, the protozoa *Cryptosporidium* spp. and *Giardia duodenalis* are major contributors to the burden of diarrhoeal disease globally [2]. *Cryptosporidium* spp., with *C. parvum* and *C. hominis* accounting for most documented cases of human cryptosporidiosis, have been recognised as the cause of large water- and food-borne outbreaks of gastroenteritis [3,4,5]. *Giardia duodenalis* infections are very common throughout the world and are regarded as a major cause of gastrointestinal complaints in western countries [6]. Another intestinal protozoan with a cosmopolitan distribution, although with highly variable colonisation/infection rates, is *Dientamoeba fragilis*. Although its pathogenicity and clinical significance are still controversial, most clinicians agree that *D. fragilis* can cause abdominal pain and diarrhoea as it is frequently found in patients suffering from these disorders [7]. Remarkably, the perception of the clinical significance of *D. fragilis* differs greatly among geographical areas. This protozoon is a common finding in Danish children attending day-care centres and, therefore, is regarded as a commensal organism [8]. In contrast, *D. fragilis* is less prevalent, but more frequently associated with gastrointestinal manifestations, in other high-income countries such as Australia [9]. These data suggest that implementing routine (molecular) screening of *D. fragilis* might be useful in those parts of the world where the parasite has received little attention to gain information on prevalence across various age groups and phenotypes (symptomatic vs. asymptomatic). This information would be extremely useful to help in ascertaining the true clinical significance of *D. fragilis*.

Classically, diagnosis of protozoan enteroparasite infections has been achieved by microscopical examination of faecal samples. Light microscopy of faecal concentrates stands as the preferred routine diagnostic method in many clinical settings. Although this technique is labour-intensive, lacks sensitivity and requires skilled technicians, its simplicity and low cost outweighs the above-mentioned limitations and makes microscopy suited for resource-limited laboratories, particularly those in endemic and high-prevalence areas [10]. In contrast, typical epidemiological scenarios in high-income countries involve low parasite prevalence rates and burdens. In this context, highly sensitive and specific diagnostic methods are required [11,12]. Accurate and fast diagnosis of intestinal protozoa is also important in treatment decision-making schemes, as well as for early detection in outbreak investigations. Not surprisingly, molecular techniques are progressively replacing microscopy as the first-line diagnostic method for diagnosis of intestinal parasites in industrialised countries [12].

In recent years, a wide diversity of in-house and commercial real-time PCR (qPCR) assays (see Appendix A) have been developed for detecting enteric parasitic pathogens, with the trend moving quickly from single to multiple pathogen detection [6,13]. Validation and standardisation of such novel assays are important tasks that must be carried out to accurately assess their diagnostic performance and practical suitability for routine use in clinical settings. Here, we describe the development, optimisation and validation of a fast and practical in-house multiplex qPCR for the simultaneous detection of *Cryptosporidium* spp., *Giardia duodenalis* and *Dientamoeba fragilis* in human stool samples.

## 2. Materials and Methods

### 2.1. Study Design

This was a retrospective observational study specifically conducted to develop, validate and standardise a novel multiplex qPCR method for the simultaneous detection of the diarrheagenic protozoan enteroparasites *Cryptosporidium* spp., *G. duodenalis* and *D. fragilis*.

### 2.2. DNA Reference Panel

A total of 424 well-characterised genomic DNA samples extracted and purified from biological (mostly stool) specimens of clinical patients were used (Table 1). These samples were initially diagnosed by in-house singleplex PCR protocols and, when possible, Sanger sequencing at reception at the Spanish National Centre for Microbiology (SNCM). DNA samples positive for *Cryptosporidium* spp. (*n* = 126; median cycle threshold (C_T_) value, 32.6; range, 25.5–40.0) and *Giardia duodenalis* (*n* = 132; median C_T_ value, 31.0; range, 24.1–40.0) were initially detected by a commercial multiplex qPCR platform (Allplex™ Gastrointestinal-Parasite Assay, Seegene Inc., Seoul, South Korea). DNA samples positive for *D. fragilis* (*n* = 49) had initially been identified by an in-house singleplex qPCR (median C_T_ value, 32.2; range, 21.4–41.3) [14]. DNA samples (*n* = 105) positive for enteric or phylogenetically related (e.g., Apicomplexan) parasitic species other than *G. duodenalis* and *D. fragilis* and those belonging to *Cryptosporidium* (and used in cross-reaction evaluation in the present study) were initially diagnosed by a variety of in-house single-round, semi-nested and nested PCR assays in place at the SNCM. DNA (*n* = 12) from stool samples of apparently healthy individuals was included for the same purpose. Whenever possible, species and genotypes/sub-genotypes were determined by PCR and Sanger sequencing methods at the time of initial diagnosis. The full dataset including all relevant information of the original DNA samples used in this study can be found in Appendix A.

The DNA reference panel included DNA samples of the four *Cryptosporidium* species most frequently found causing human cryptosporidiosis (*C. hominis*, *C. parvum*, *C. meleagridis* and *C. cuniculus*) [15,16] and of five different *G. duodenalis* genetic variants (assemblages A–E). For cross-reactivity evaluation, we purposely selected representative DNA samples of various taxonomic groups of pathogens, including amoebozoan (*n* = 38), apicomplexan (*n* = 41), euglenozoan (*n* = 13), microsporidian (*n* = 3), nematode (*n* = 8) and platyhelminthic (*n* = 2) parasites (Table 1).

All DNA samples (except those positive for *D. fragilis*) were part of the DNA collection of the Parasitology Reference and Research Laboratory of the SNCM. Genomic DNA was extracted from 200 mg of faecal material using the QIAamp DNA Stool Mini Kit (QIAGEN, Hilden, Germany) according to the manufacturer’s instructions. Some DNA samples were from in vitro cultures or of animal origin, particularly those belonging to animal-adapted species/genotypes or rarely found circulating in humans (Appendix A). DNA samples from *D. fragilis*-positive patients were provided by the Department of Bacteria, Parasites and Fungi, Infectious Disease Preparedness, Statens Serum Institute (Copenhagen, Denmark). All DNA samples used in this study had been extracted during the period 2014–2019 and stored at –20 °C until testing.

### 2.3. Primers and Probes

PCR primers and detection probes were chosen using Primer Express software (Applied Biosystems, CA, USA) based on the known oocyst wall protein 1 (*cowp1*) gene sequence for *C. hominis* (GenBank accession No. DQ388389), the small subunit ribosomal RNA (*ssu* rRNA) gene sequence for *G. duodenalis* (GenBank accession No. M54878) and the internal transcriber spacer (ITS) gene sequence for *D. fragilis* (KY554844). Selected primer and probe sequences are detailed in Table 2. The *C. hominis*-specific (forward and reverse) primers and probe set comprised positions 237–261, 387–365 and 356–326 of DQ388389, respectively. The *G. duodenalis*-specific (forward and reverse) primers and probe set comprised positions 79–98, 141–126 and 105–124 of M54878, respectively. The *D. fragilis*-specific (forward and reverse) primers and probe set comprised positions 96–117, 193–169 and 169–144 of KY554844, respectively.

### 2.4. Optimisation

Singleplex or multiplex qPCR reactions were carried out in volumes of 25 µL with PCR buffer (2× Brilliant III Ultra-Fast QPCR master mix, Agilent Technologies, Santa Clara, CA, USA) containing mutant Taq DNA polymerase, dNTPs and 5.5 mM of MgCl_2_. Mixes also included 50 pmol of each specific primer, 25 pmol of double-labelled probes and 5 µL of template DNA. Cycling conditions consisted of 10 min incubation at 95 °C, followed by 45 cycles of 15 s at 95 °C and 1 min at 60 °C, with fluorescence being measured after each cycle. Amplification was performed with the BioRad CFX96 Dx System (BioRad, Hercules, CA, USA).

### 2.5. Performance Assessment

All samples forming part of the DNA reference panel were blindly analysed in duplicates to avoid bias. Samples positive for *Cryptosporidium* spp. (ROX fluorophore), *G. duodenalis* (FAM fluorophore) and *D. fragilis* (HEX fluorophore) typically generated fluorescence curves with qPCR C_T_ values < 45.

### 2.6. Limit of Detection and PCR Efficiency

Ten-fold serial dilutions (down to 10^−7^) of individual DNA samples positive for *C. hominis*, *G. duodenalis* and *D. fragilis* were used to estimate the relative limit of detection of the qPCR assay in both singleplex and multiplex formats. DNA isolated from 10,000 *C. hominis* oocysts and 500 *G. duodenalis* cysts (enumerated from human faecal samples using direct fluorescence microscopy) were used as starting concentrations for this purpose. No quantified starting material was available for *D. fragilis*, as cyst forms from this protozoon have not been unambiguously described in human clinical specimens [17]. Additionally, the efficiency (E) of the multiplex qPCR assay was calculated from the slope(s) of the standard curve according to the following formulas: E = 10(−1/slope) −1(1)
Log E = (−1/slope)log 10−log 1(2)
Log 2 = (−1/slope) × 1−0 (because E = 2, log 1 = 0 and log10 = 1)(3)
Slope= −1/log2 (after multiplying both sides by (slope/log2)(4)
Slope = −3.32(5)

Therefore, for a 10-fold dilution series, the slope is −3.33 when E = 100%. A good reaction should have an efficiency between 90% and 110%, which corresponds to a slope between −3.58 and −3.10 [18].

### 2.7. Statistical Analyses

The Cohen’s Kappa test was estimated to assess the agreement of the diagnostic results obtained with the multiplex qPCR detection assay and the reference singleplex qPCR methods used during routine analyses at initial diagnosis (see Appendix A). Cohen’s Kappa ranges between 0 (no agreement between the two raters) and 1 (perfect agreement between the two raters). A Cohen’s Kappa value between 0.81 and 0.99 was considered as “near-perfect agreement”. Clinical diagnostic sensitivity and specificity, and negative and positive predicted values (with 95% confidence intervals), were calculated using Meta-DiSc 1.4. software [19] based on the following formulas:Sensitivity (Se) = [a/(a + c)] × 100(6)
Specificity (Sp) = [d/(b + d)] × 100(7)
Positive predictive value (PPV) = [a/(a + b)] × 100(8)
Negative predictive value (NPV) = [d/(c + d)] × 100(9)
where a = true-positive samples, b = false-positive samples, c = false-negative samples and d = true-negative samples. C_T_ values ≥ 36 with a simultaneous baseline-corrected normalised fluorescence value < 1500 were considered doubtful results. Reference DNA samples positive for *Cryptosporidium*, *G. duodenalis* and *D. fragilis* that yielded a negative or a doubtful result in the novel multiplex assay were re-assessed by singleplex qPCR. DNA samples with a negative result in the novel multiplex assay but a positive result in the subsequent confirmatory singleplex qPCR were considered true false negatives.

## 3. Results

### 3.1. Limit of Detection and Efficiency

The analytical sensitivity and efficiency of the qPCR assay for the detection and differentiation of *Cryptosporidium* spp., *G. duodenalis* and *D. fragilis* (both in singleplex and multiplex formats) are summarised in Table 3. The assay was able to detect the equivalent of a single *Cryptosporidium* oocyst in both the singleplex and multiplex formats. However, the latter performed slightly poorer than the former in terms of efficiency (110% vs. 100.3%, respectively). These results are consistent with the facts that: (i) a single *Cryptosporidium* oocyst contains four sporozoites, each with a haploid genome, resulting in an oocyst ploidy of 4N, and its *cowp1* gene exists in a single copy, (ii) a fully differentiated *Giardia* cyst contains four nuclei, each with a ploidy of 4N, resulting in a cyst ploidy of 16N, and its *ssu* rRNA gene exists in multiple copies, and (iii) *Dientamoeba* is unknown with respect to ploidy and genome size, but the targeted gene (ITS) should be multi-copy, one for each ribosomal set of genes.

The multiplex format of the assay was particularly sensitive for the detection of *G. duodenalis* (limit of detection: 5 × 10^−4^ cysts) and *D. fragilis* (limit of detection: dilution 1 × 10^−7^). The multiplex format was more efficient than the equivalent singleplex formats in detecting *G. duodenalis* (98.8% vs. 86.9, respectively) and *D. fragilis* (101.1% vs. 98.6%, respectively).

### 3.2. Sensitivity and Specificity

The diagnostic performance of the multiplex qPCR assay is summarised in Table 4. Out of the 126 *Cryptosporidium*-positive samples included in the DNA reference panel, the multiplex assay was able to detect 119 (diagnostic sensitivity, 94.4%). Samples belonged to *C. hominis* (*gp60* genotype families Ib, Id, Ie and If), *C. parvum* (*gp60* genotype families IIa and IId), *C. meleagridis* (*gp60* genotype family IIIg) and *C. ubiquitum* (unknown *gp60* genotype family) (see Appendix A). The seven *Cryptosporidium*-positive samples that failed to be amplified in the multiplex qPCR assay included one *C. hominis*, one *C. parvum*, three *C. meleagridis* and two *Cryptosporidium* spp. isolates. All seven *Cryptosporidium*-positive samples yielded C_T_ values > 35 when retested in singleplex qPCR and were, therefore, regarded as true false-negative results in multiplex qPCR. Of note, 37 *Cryptosporidium*-positive samples were known to be concomitantly infected by *D. fragilis* (*n* = 26), *G. duodenalis* (*n* = 10) and *D. fragilis* + *G. duodenalis* (*n* = 1), as previously determined during routine initial diagnosis. All these samples were also scored positive for *G. duodenalis* and *D. fragilis* by the multiplex qPCR assay and, therefore, not considered as false-positive results (Appendix A).

Out of the 132 *G. duodenalis*-positive samples included in the DNA reference panel, the multiplex assay was able to detect 129 (diagnostic sensitivity, 97.7%). Samples belonged to *G. duodenalis* assemblages A, B, C, D and E (see Appendix A). The three *Giardia*-positive samples that failed to be amplified in the multiplex assay included two assemblage B isolates, whereas the assemblage type of the third isolate was unknown. The multiplex assay also detected co-infections by *D. fragilis* (*n* = 50) and *D. fragilis* + *Cryptosporidium* spp. (*n* = 1), all of them previously identified during routine initial diagnosis (Appendix A).

Out of the 49 *D. fragilis*-positive samples included in the DNA reference panel, the assay was able to detect 44 (diagnostic sensitivity: 89.8%). No attempts were made to determine the genotype of the *D. fragilis* isolates because of the limited intra-genetic diversity previously observed in this protozoon [20,21]. A total of 18 *D. fragilis*-positive samples were co-infected either with *G. duodenalis* (*n* = 17) or *Cryptosporidium* (*n* = 1). All these concomitant infections were identified at initial diagnosis and therefore were not considered as false-positive results (Appendix A). To discard potential cross-reactivity with phylogenetically closely related members of the genus *Trichomonas*, we aligned our *D. fragilis* sequence primers (both forward and reverse) with known partial *ssu* rRNA sequences from *T. vaginalis* (GenBank accession number JX943584), *T. gallinae* (GenBank accession number EU215373) and *T. tenax* (GenBank accession number D49495). In all cases, mismatches were found in 36–41% of the positions, making the amplification of *Trichomonas* spp. with our *D. fragilis* primer set highly unlikely.

None of the 105 DNA samples positive for parasites other than *Cryptosporidium* spp., *G. duodenalis* and *D. fragilis* yielded positive results when tested in the multiplex assay, nor did the 12 DNA samples from stool samples of individuals without clinical manifestations produce any signal (Table 4).

Finally, a very good agreement (Kappa test values ranging from 0.925 to 0.976) was observed between the results obtained by multiplex qPCR assay and those previously generated by the reference qPCR methods at initial diagnosis (Table 5).

## 4. Discussion

Here, we developed, optimised and evaluated a new multiplex qPCR assay for the simultaneous detection of the three most common and clinically relevant protozoan enteroparasites (*Cryptosporidium* spp., *G duodenalis* and *D. fragilis*) in western countries. To do so, a large, carefully selected panel of well-characterised DNA samples was used. This is the strategy adopted in our laboratory for evaluating and validating PCR-based diagnostic tests [22,23]. This approach has two additional benefits, as it allows (i) the testing of infrequent species/genotypes and animal-adapted genetic variants with zoonotic potential, and (ii) the testing of species/genetic variants from two European countries (Denmark and Spain), reflecting different geographic areas and epidemiological scenarios. In contrast, most previous studies preferred using prospectively collected stool samples submitted to clinical laboratories for parasite investigation, for which the above-mentioned molecular information is often unavailable [24,25,26,27,28,29,30,31,32].

Our multiplex qPCR assay achieved a diagnostic sensitivity of 0.94 for the detection of *Cryptosporidium* spp. This figure is slightly lower than those (ca. 1) reported using commercially available multiplex qPCR assays, including the Allplex GI-Parasite (Seegene, Soul, South Korea) [33,34], the EasyScreen Enteric Parasite (Genetic Signatures, Sydney, Australia) [27], the G-DiaParaTrio (Diagenode Diagnostics, Liège, Belgium) [34] and the NanoCHIP (Savyon Diagnostics, Ashdod, Israel) [30] assays. The slightly lower sensitivity of our multiplex qPCR assay is likely due to the single-copy nature of the targeted gene (*cowp1*) compared to the multiple-copy nature of *ssu* rRNA and ITS. In contrast, *cowp1* has increased specificity over *ssu* rRNA, reducing the rate of false-positive amplifications. Still, lower values of 0.86–0.92 have been reported with the RIDAGENE Parasitic Stool Panel II (R-Biopharm AG, Darmstadt, Germany) [33,34] and the Tib MolBiol (Roche Diagnostics, Basilea, Switzerland) [33] assays. Remarkably, our multiplex qPCR assay was able to detect the four *Cryptosporidium* species (namely *C. hominis*, *C. parvum*, *C. meleagridis* and *C. ubiquitum*) causing the bulk of human cryptosporidiosis cases globally.

Regarding *G. duodenalis*, our multiplex qPCR test achieved a diagnostic sensitivity of 0.97, a figure in the lower range of those (0.97–1) reported by the best commercially available assays. These include the Allplex (Seegene) [33,34], the Tib MolBiol (Roche Diagnostics) [33] or the NanoCHIP (Savyon Diagnostics) [26,30] methods. Poorer performances (≤0.92) have been documented for the EasyScreen (Genetic Signatures), the FTD Stool Parasites (Fast Track Diagnostics, Esch-sur-Alzette, Luxemburg) and the G-DiaParaTrio (Diagenode Diagnostics) [22,27,31,33,34,35] assays.

In many western countries, the trichomonadida parasite *D. fragilis* is a very common finding in human stool samples and is second only to *Blastocystis* sp. in terms of positive rates [36]. Since *D. fragilis* is particularly difficult to identity using conventional microscopy examination [37], qPCR has become the gold standard for the diagnosis of this (and other) gastrointestinal protozoan parasites. Indeed, several qPCR methodologies for the simultaneous detection of diarrhoea-causing protozoan pathogens, including *D. fragilis,* have been developed and commercialised in the last decade (see Appendix A). A recent study documented that discrepant results in the detection of *D. fragilis* were common when the diagnostic performances of laboratory-developed and commercial qPCR assays were compared [38]. The multiplex qPCR assay described here achieved a diagnostic sensitivity of 0.90 for the detection of *D. fragilis*, well in the range of those reported in the literature using commercial methods (see Appendix A). Higher sensitivity values (0.95–1) have been reported using the Allplex (Seegene) [33,34], EasyScreen (Genetic Signatures) [27], FTD (Fast Track Diagnostics) [22,25,33,35], NanoCHIP (Savyon Diagnostics) [26,30] and the Tib MolBiol (Roche Diagnostics) methods [33]. Lower (0.25–0.79) diagnostic sensitivities have been reported using RIDAGENE (R-Biopharm) [33,34]

Regardless of the pathogen considered, our multiplex qPCR assay produced diagnostic specificities of 1. This was somehow expected as the panel used for reference purposes included well-characterised DNA samples that were previously diagnosed for the potential presence of concomitant infections, minimizing the risk of obtaining false-positive results. Despite this effort, we are aware that some relevant pathogenic and commensal protozoan species were missing in our panel. For instance, DNA samples from host-adapted *Cryptosporidium* species rarely seen infecting humans (e.g., *C. canis*, *C. felis*, *C. muris*) were not available for testing. Other potentially cross-reacting species that were not part of the reference panel included *Cyclospora cayetanensis*, *Entamoeba coli*, *Endolimax nana* and *Encephalitozoon intestinalis*, among others. Similarly, larger numbers of *G. duodenalis* assemblages C, D and E should be tested to confirm the diagnostic sensitivity achieved for isolates belonging to assemblages A and B. These limitations remain to be addressed in future works.

## 5. Conclusions

The novel multiplex qPCR test described here exhibited excellent linearity and high analytical sensitivity and specificity values, providing a suitable alternative diagnostic tool for the molecular detection and differentiation of *Cryptosporidium* spp., *G. duodenalis* and *D. fragilis*, the most clinically relevant diarrhoea-causing protozoan parasites in high-income countries. The test can also be used either for screening or confirmatory purposes in endemic areas of resource-poor settings. Of note, detection of all three pathogens (but especially *D. fragilis*) by conventional microscopy examination is challenging because of the intrinsic difficulty of correctly identifying some of their morphological features or the need for specific staining methods (e.g., Ziehl–Neelsen staining for *Cryptosporidium* spp.) [39,40].

## Figures and Tables

**Table 1 pathogens-11-01277-t001:** Panel of laboratory-confirmed DNA samples used in the present study.

Phylum	Genus	Species	DNA Samples (*n*)
Apicomplexa	*Cryptosporidium*	*C. cuniculus*	1
		*C. hominis*	72
		*C. meleagridis*	4
		*C. parvum*	30
		*Cryptosporidium* spp. ^1^	19
Metamonada	*Giardia*	*G. duodenalis*	132
	*Dientamoeba*	*D. fragilis*	49
Amoebozoa	*Entamoeba*	*E. histolytica*	27
		*E. dispar*	11
Apicomplexa	*Babesia*	*B. divergens*	1
	*Besnoitia*	*B. besnoiti*	4
		*B. tarandi*	1
	*Neospora*	*N. caninum*	5
	*Plasmodium*	*P. falciparum*	13
		*P. malariae*	1
		*P. ovale*	4
		*P. vivax*	2
	*Sarcocystis*	*S. tenella*	5
	*Toxoplasma*	*T. gondii*	5
Euglenozoa	*Leishmania*	*L. aethiopica*	1
		*L. amazonensis*	1
		*L. braziliensis*	1
		*L. donovani*	1
		*L. infantum*	1
		*L. major*	1
		*L. mexicana*	1
		*L. tropica*	1
	*Trypanosoma*	*T. cruzi*	5
Microsporidia	*Enterocytozoon*	*E. bieneusi*	3
Nematoda	*Ancylostoma*	*A. duodenale*	2
	*Ascaris*	*A. lumbricoides*	2
	*Necator*	*N. americanus*	1
	*Trichuris*	*T. muris*	1
	*Strongyloides*	*S. venezuelensis*	2
Platyhelminthes	*Taenia*	*T. saginata*	1
		*T. solium*	1
Sub-total			412
Healthy patients	Uninfected	Uninfected	12
Total			424

^1^ Sanger sequences of insufficient quality to unambiguously ascertain the *Cryptosporidium* species involved.

**Table 2 pathogens-11-01277-t002:** Primers and probes used in the multiplex real-time PCR assay for the simultaneous detection of *Cryptosporidium* spp., *Giardia duodenalis* and *Dientamoeba fragilis* in the present study.

Target Organism	Target Gene	Forward Primer (5′-3′)	Reverse Primer (5′-3′)	Probe (5′-3′)
*Cryptosporidium* spp.	*cowp1*	CAAATTGATACCGTTTGTCCTTCTG	GGCATGTCGATTCTAATTCAGCT	ROX–TGCCATACATTGTTGTCCTGACAAATTGAAT–BHQ
*Giardia duodenalis*	*ssu* rRNA	GACGGCTCAGGACAACGGTT	TTGCCAGCGGTGTCCG	FAM–CCCGCGGCGGTCCCTGCTAG–BHQ1
*Dientamoeba fragilis*	ITS	CAACGGATGTCTTGGCTCTTTA	TGCATTCAAAGATCGAACTTATCAC	HEX–CAATTCTAGCCGCTTAT–MGB
Internal control	Synthetic ^1^	ACATTCGCAACATACGCCATACT	TAAGACAGCTGCTACAGGCACACT	Cyanine5–ATCCGCGCACCTGCCGTCTCTA–BHQ2

*cowp1*, *Cryptosporidium* oocyst wall protein 1; ITS, internal transcribed spacer; *ssu* rRNA, small subunit ribosomal RNA. ^1^ Artificial template.

**Table 3 pathogens-11-01277-t003:** Analytical sensitivity and efficiency of the multiplex real-time PCR assay for the simultaneous detection of *Cryptosporidium* spp., *Giardia duodenalis* and *Dientamoeba fragilis*.

Target Organism	qPCR Format	Variable	Real-Time PCR Cycle Threshold (C_T_) Values by (oo)cyst Number or DNA Concentration								R^2^	Slope	Efficiency (%)
*Cryptosporidium* spp.		Oocysts (*n*)	1 × 10^4^	1 × 10^3^	1 × 10^2^	1 × 10^1^	1	1 × 10^−1^	1 × 10^−2^	1 × 10^−3^			
	Singleplex		24.07	27.22	30.25	33.47	37.52	Neg.	Neg.	Neg.	0.990	–3.31	100.3
	Multiplex		24.17	27.56	30.54	33.73	36.55	Neg.	Neg.	Neg.	0.999	–3.10	110
*Giardia duodenalis*		Cysts (*n*)	5 × 10^2^	5 × 10^1^	5	5 × 10^−1^	5 × 10^−2^	5 × 10^−3^	5 × 10^−4^	5×10^−5^			
	Singleplex		22.31	25.93	28.99	33.56	Neg.	Neg.	Neg.	Neg.	0.994	–3.68	86.9
	Multiplex		22.81	26.37	29.57	32.72	36.39	36.80	37.72	Neg.	0.999	–3.35	98.8
*Dientamoeba fragilis*		[DNA]	Neat	1 × 10^−1^	1 × 10^−2^	1 × 10^−3^	1 × 10^−4^	1 × 10^−5^	1 × 10^−6^	1 × 10^−7^			
	Singleplex		21.74	25.29	28.47	31.81	35.26	35.72	37.25	36.70	1	–3.35	98.6
	Multiplex		22.14	25.34	28.58	31.85	35.18	38.64	37.63	Neg.	1	–3.29	101.1

Neg., Negative result. Regression coefficients (R^2^), slopes and efficiency values are also indicated.

**Table 4 pathogens-11-01277-t004:** Diagnostic performance of the multiplex real-time PCR assay for the simultaneous detection of *Cryptosporidium* spp., *Giardia duodenalis* and *Dientamoeba fragilis* against the selected DNA reference panel. 95% confidence intervals (CI) are indicated.

Target organism	Accuracy ^1^ (95% CI)	TP	TN	FP	FN	Sensitivity (95% CI)	Specificity (95% CI)	Positive Predictive Value	Negative Predictive Value (95% CI)
*Cryptosporidium* spp.	0.97 (0.94–0.99)	126	119	0	7	0.94 (0.89–0.98)	1.00 (0.96–1.00)	1.00	0.94 (0.89–0.97)
*Giardia duodenalis*	0.99 (0.96–0.99)	132	129	0	3	0.97 (0.94–0.99)	1.00 (0.97–1.00)	1.00	0.97 (0.93–0.99)
*Dientamoeba fragilis*	0.97 (0.93–0.99)	49	44	0	5	0.90 (0.80–0.97)	1.00 (0.97–1.00)	1.00	0.96 (0.91–0.98)
All three	0.97 (0.94–0.98)	307	292	0	15	0.95 (0.92–0.97)	1.00 (0.97–1.00)	1.00	0.89 (0.83–0.93)

FP, false positive; FN, false negative; TN, true negative; TP, true positive. ^1^ Overall probability that a sample is correctly classified. This value is dependent on disease prevalence.

**Table 5 pathogens-11-01277-t005:** Direct performance comparison of the multiplex real-time PCR assay for the simultaneous detection of *Cryptosporidium* spp., *Giardia duodenalis* and *Dientamoeba fragilis* with reference to singleplex real-time PCR methods used during routine analyses at initial diagnosis.

Protozoan Species	(mqPCR+/sqPCR+)	(mqPCR–/sqPCR+)	(mqPCR+/sqPCR–)	(mqPCR–/sqPCR–)	Kappa Test	% Agreement
*Cryptosporidium* spp.	119	7	0	117	0.942	97.1
*Giardia duodenalis*	129	3	0	117	0.976	98.8
*Dientamoeba fragilis*	44	5	0	117	0.925	96.9

mqPCR: multiplex real-time PCR; sqPCR: singleplex real-time PCR.

## Data Availability

Data are available within the article or its Appendix A.

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
