# Peer review of "Development, Optimisation and Validation of a Novel Multiplex Real-Time PCR Method for the Simultaneous Detection of Cryptosporidium spp., Giardia duodenalis and Dientamoeba fragilis"

_pathogens, 2022, doi:10.3390/pathogens11111277_

Round 1

Reviewer 1 Report

This is a very nice study that generated single and multiplex qPCR assays, with validation, for the detection of the commonly occurring protist pathogens, Cryptosporidium (many common human-infecting species), Giardia (multiple assemblages) and the trichomonad, Dientamoeba fragilis.  The assay is a probe-based Taq-man assay (this adds additional specificity and signal) utilizing three different fluorophores (ROX, FAM, HEX) and the gene targets are cowp in Cryptosporidium, the multi-copy SSUrRNA in Giardia and the multi-copy ITS in Dientamoeba fragilis.  A large number of confounders (other protists and parasites) and controls were tested and I am satisfied, excepting one significant comment below that the method is sound and that this assay will be of use in countries and laboratories that can afford molecular diagnostics.

1) In my experience, the largest obstacle in developing globally applicable diagnostics is the identification of targets that are sufficiently stable across global populations.  Line 275 argues that the assays here were performed on samples from diverse geographic areas, however no data regarding the geographic origins of the samples is provided. Please add this information to the sample table(s) and elaborate on the number of continents and countries represented. Even if the assay is only proven in certain regions (excepting travelers) it should be of value.

2) There are multiple oocyst wall proteins encoded by the Cryptosporidium genome sequence. Exactly which gene, by gene ID was used? It looks like it might have been cowp1 based on searches of CryptoDB.org but this should be verified. As this gene has been the subject of multiple population studies it should be possible to assess the conservation of the regions used for the primer and probe design against existing sequences effectively augmenting your sample pool and geographic diversity.

3) Likewise, the sequences for other coccidian pathogens for which the authors did not have samples (e.g. Trichomonas vaginalis comes to mind as I see it was not listed and it would be the closest taxonomically to compare with Dientamoeba fragilis), can be searched to see if the primers and probe will bind well to their oocyst wall genes using bioinformatics. The gold standard will always be performance in the assay but insights can be drawn from the sequences as well.

4) Perhaps this is minor, but I was struck by the emphasis on high-income in the abstract and western countries line 270.  The pathogens exert such a high burden globally, especially in low income countries I was a bit taken aback. It is true that these are protist pathogens that are income blind and hence truly global including higher-income countries. It would be nice to find a way to word this in a way that acknowledges the true burden of these pathogens and that for laboratories and countries equipped for molecular diagnostics, this assay may prove useful.

Minor issues:

1 - Add a reference for lacks sensitivity to line 70

2 - Perhaps change fast to quickly, line 82

3 - Table 3 - if gene copy numbers are considered a bit further, the Table makes even better sense.  A single Cryptosporidium oocysts will contain 4 haploid genome sequences so, 4N copies of a single-copy gene.  A Giardia mature cyst is 16N and the SSUrRNA exists in multiple copies. Dientamoeba is unknown with respect to ploidy and genome size, but the target, ITS should be multi-copy, one for each ribosomal set of genes and the limiting dilution experiments support this given their sensitivity. 

4 - Table 5 column headers are a little confusing as the title says comparison of multiplex to single but the presentation suggests, single/multi. It might be nice to clarify the column headers.  

5 - Both the SSUrRNA and ITS are multi-copy, line 286

Author Response

Reviewer #1

This is a very nice study that generated single and multiplex qPCR assays, with validation, for the detection of the commonly occurring protist pathogens, Cryptosporidium (many common human-infecting species), Giardia (multiple assemblages) and the trichomonad, Dientamoeba fragilis. The assay is a probe-based Taq-man assay (this adds additional specificity and signal) utilizing three different fluorophores (ROX, FAM, HEX) and the gene targets are cowp in Cryptosporidium, the multi-copy SSU rRNA in Giardia and the multi-copy ITS in Dientamoeba fragilis. A large number of confounders (other protists and parasites) and controls were tested and I am satisfied, excepting one significant comment below that the method is sound and that this assay will be of use in countries and laboratories that can afford molecular diagnostics.

Reply: we thank Reviewer #1 for his/her preliminary positive appraisal. We have considered all his/her comments and suggestions and reply to them (see below) as thoroughly as possible. Changes introduced in the text have been highlighted in red for better identification.

  • In my experience, the largest obstacle in developing globally applicable diagnostics is the identification of targets that are sufficiently stable across global populations. Line 275 argues that the assays here were performed on samples from diverse geographic areas, however no data regarding the geographic origins of the samples is provided. Please add this information to the sample table(s) and elaborate on the number of continents and countries represented. Even if the assay is only proven in certain regions (excepting travellers) it should be of value.

Reply: Following Reviewer #1 advice, we have included in Table S2 information on the geographical origin (country) of the DNA samples used in the present study. All D. fragilis-positive samples were from Danish patients as commented in current lines 129-131. The remaining DNA samples were from Spain. This information has been now clarified in Table S2 and current lines 288-289 of the Discussion section.

  • There are multiple oocyst wall proteins encoded by the Cryptosporidium genome sequence. Exactly which gene, by gene ID was used? It looks like it might have been cowp1 based on searches of CryptoDB.org but this should be verified. As this gene has been the subject of multiple population studies it should be possible to assess the conservation of the regions used for the primer and probe design against existing sequences effectively augmenting your sample pool and geographic diversity.

Reply: Reviewer #1 is right, cowp1 was the targeted gene for detecting Cryptosporidium spp.in the present study. This issue has been now clarified in current lines 135 and 300 and in Table 2.

  • Likewise, the sequences for other coccidian pathogens for which the authors did not have samples (e.g. Trichomonas vaginalis comes to mind as I see it was not listed and it would be the closest taxonomically to compare with Dientamoeba fragilis), can be searched to see if the primers and probe will bind well to their oocyst wall genes using bioinformatics. The gold standard will always be performance in the assay but insights can be drawn from the sequences as well.

Reply: Reviewer #1 raised an interesting point. We did not include T. vaginalis in our panel of potential cross-reacting pathogens because no DNA from this parasite was available in our lab for this specific purpose. Following Reviewer #1 advice, we aligned our D. fragilis sequence primers (both forward and reverse) with known partial 18S sequences for T. vaginalis (GenBank accession number JX943584), T. gallinae (GenBank accession number EU215373) and T. tenax (GenBank accession number D49495). In all cases mismatches were found in 36-41% of the positions, making highly unlikely the amplification of Trichomonas spp. with this primer set. This information has been now added in current lines 262-268 of the Results section.

  • Perhaps this is minor, but I was struck by the emphasis on high-income in the abstract and western countries line 270. The pathogens exert such a high burden globally, especially in low income countries I was a bit taken aback. It is true that these are protist pathogens that are income blind and hence truly global including higher-income countries. It would be nice to find a way to word this in a way that acknowledges the true burden of these pathogens and that for laboratories and countries equipped for molecular diagnostics, this assay may prove useful.

Reply: We agree with Reviewer #1 with the fact that all Cryptosporidium spp., Giardia duodenalis, and Dientamoeba fragilis parasites are ubiquitous and are major contributors (particularly the former) to the global diarrhoea burden particularly in poor-resource settings. We wanted to highlight the convenience of multiplex qPCR methods in medium- and high-income countries because in these settings prevalence rates are usually lower and in these epidemiological scenarios diagnostic sensitivities become an issue. This of course does not preclude that the test can be also used in highly endemic areas. We have acknowledged this possibility in lines 339-341 of the Conclusions section.

  • Add a reference for lacks sensitivity to line 70

Reply: the reference Friesen et al Clin Microbiol Infect. 2018;24:1333–1337. doi: 10.1016/j.cmi.2018.03.025. has been added in current lines 73 and 398-400 to illustrate the reduced diagnostic sensitivity of microscopy when compared with multiplex qPCR platforms. The reference was chosen because the study included the same enteric pathogen investigated in ours and was tested against a large panel (<1000) of faecal DNA samples. All subsequent references quoted in the main body of the manuscript have been renumbered accordingly.

  • Perhaps change fast to quickly, line 82

Reply: The term “fast” has been replaced by the term “quickly” as recommended by Reviewer #1.

  • Table 3 - if gene copy numbers are considered a bit further, the Table makes even better sense. A single Cryptosporidium oocysts will contain 4 haploid genome sequences so, 4N copies of a single-copy gene. A Giardia mature cyst is 16N and the SSU rRNA exists in multiple copies. Dientamoeba is unknown with respect to ploidy and genome size, but the target, ITS should be multi-copy, one for each ribosomal set of genes and the limiting dilution experiments support this given their sensitivity.

Reply: We thank Reviewer #1 for this useful information, which has been added to the manuscript in current lines 209-215 of the Results section.

  • Table 5 column headers are a little confusing as the title says comparison of multiplex to single but the presentation suggests, single/multi. It might be nice to clarify the column headers.

Reply: Table 5 column headers have been modified to improve clarity as recommended by Reviewer #1.

  • Both the SSUrRNA and ITS are multi-copy, line 286

Reply: Information updated as per requested in current line 300 of the Discussion section.

Reviewer 2 Report

This is well written and useful article. However, I have one specific comment and one doubt.

Can the authors add any details regarding extraction of DNA from fecal samples? It was shown that the method of DNA isolation from fecal samples have an influence on PCR sensitivity (doi: 10.1128/JCM.01823-06)

I am not sure if it is proper to write that in case of Giardia the method evaluated by the authors has 97.7 % sensitivity and that the method allow to detect Giardia assemblages A, B, C, D, and E. There were only two samples of C assemblage, one sample of D, and one sample of E assemblage (Table S2). Many readers do not read supplementary files. Thus, the readers of the article may conclude from the article that there is high sensitivity (97.7 %) of the method described in this manuscript in case of assemblages C, D and E, but it has not been proved here. Moreover, it may be concluded that this method is highly sensitive in veterinary practice, although only 3 positive samples were from dogs, and only one from cattle. This should be emphasized in the discussion section. Or the animal samples should be excluded from the study.

Author Response

Reviewer #2

This is well written and useful article. However, I have one specific comment and one doubt.

Reply: we thank Reviewer #2 for his/her preliminary positive appraisal. We have considered all his/her comments and suggestions and reply to them (see below) as thoroughly as possible. Changes introduced in the text have been highlighted in red for better identification.

  1. Can the authors add any details regarding extraction of DNA from fecal samples? It was shown that the method of DNA isolation from fecal samples have an influence on PCR sensitivity (doi: 10.1128/JCM.01823-06).

Reply: All faecal DNA samples were obtained from 200 mg of faecal material using the QIAamp DNA Stool Mini Kit (QIAGEN, Hilden, Germany) according to the manufacturer’s instructions. This information has been added in current lines 124-126 of the Materials and methods section.

  1. I am not sure if it is proper to write that in case of Giardia the method evaluated by the authors has 97.7 % sensitivity and that the method allow to detect Giardia assemblages A, B, C, D, and E. There were only two samples of C assemblage, one sample of D, and one sample of E assemblage (Table S2). Many readers do not read supplementary files. Thus, the readers of the article may conclude from the article that there is high sensitivity (97.7 %) of the method described in this manuscript in case of assemblages C, D and E, but it has not been proved here. Moreover, it may be concluded that this method is highly sensitive in veterinary practice, although only 3 positive samples were from dogs, and only one from cattle. This should be emphasized in the discussion section. Or the animal samples should be excluded from the study.

Reply: we agree with Reviewer #2. We included in our panel of DNA samples as many Giardia genetic variants as available in our lab. We have acknowledged this issue as a limitation of the study in current lines 339-341 of the Discussion section.

Reviewer 3 Report

The authors of this manuscript does an excellent job demonstrating the suitability of a multiplex real-time PCR method for the simultaneous detection of the three important protozooan parasites: Cryptosporidium, G. duodenalis and D. fragilis. The manuscript is well-written and I find that the methodology used was appropiate for this kind of studies. Although there is a limitation in the posible cross-reacting species not included in their panel, the authors, as one might expect, mention such limitation in the discusion.

Author Response

Reviewer #3

The author of this manuscript does an excellent job demonstrating the suitability of a multiplex real-time PCR method for the simultaneous detection of the three important protozoan parasites: Cryptosporidium, G. duodenalis and D. fragilis. The manuscript is well-written and I find that the methodology used was appropiate for this kind of studies. Although there is a limitation in the possible cross-reacting species not included in their panel, the authors, as one might expect, mention such limitation in the discussion.

Reply: we thank Reviewer #1 for his/her preliminary positive appraisal.